# Maternal Exposure to Ozone and the Risk of Birth Defects: A Time-Stratified Case-Crossover Study in Southwestern China

**DOI:** 10.3390/toxics12070519

**Published:** 2024-07-19

**Authors:** Yi Li, Chunbei Zhou, Jun Liu, Deqiang Mao, Zihao Wang, Qunying Li, Yunyun Wu, Jie Zhang, Qi Zhang

**Affiliations:** 1Chongqing Center for Disease Control and Prevention, Chongqing 400700, China; lily_2004_ok@hotmail.com (Y.L.); zhoucb0406@163.com (C.Z.); maodq@yeah.com (D.M.); sanschneider@163.com (Z.W.); lqycq023@163.com (Q.L.); wuyunyunscu@126.com (Y.W.); 2Department of Epidemiology, College of Preventive Medicine, State Key Laboratory of Trauma and Chemical Poisoning, Army Medical University (Third Military Medical University), Chongqing 400038, China; 3NHC Key Laboratory of Birth Defects and Reproductive Health, Chongqing Population and Family Planning Science and Technology Research Institute, Chongqing 400020, China; lj790717@126.com

**Keywords:** ozone, birth defect, case-crossover

## Abstract

A few studies have explored the relationship between air pollution exposure and the risk of birth defects; however, the ozone-related (O_3_) effects on preconception and first-trimester exposures are still unknown. In this time-stratified case-crossover study, conditional logistic regressions were applied to explore the associations between O_3_ exposure and the risk of birth defects in Chongqing, China, and stratified analyses were constructed to evaluate the modifiable factors. A total of 6601 cases of birth defects were diagnosed, of which 56.16% were male. O_3_ exposure was associated with an increased risk of birth defects, and the most significant estimates were observed in the first month before pregnancy: a 10 ug/m^3^ increase of O_3_ was related to an elevation of 4.2% [95% confidence interval (CI), 3.4–5.1%]. The associations between O_3_ exposure and congenital malformations and deformations of the musculoskeletal system were statistically significant during almost all exposure periods. Pregnant women with lower education and income, and from rural areas, were more susceptible to O_3_ exposure, with the strongest odds ratios (ORs) of 1.066 (95%CI, 1.046–1.087), 1.086 (95%CI, 1.034–1.140), and 1.053 (95%CI, 1.034–1.072), respectively. Our findings highlight the health risks of air pollution exposure and raise awareness of pregnant women’s vulnerability and the susceptibility window period.

## 1. Introduction

Air pollution is a serious environmental problem accounting for 6.7 million deaths in 2019 around the world [1]. In addition to particulate matter (PM), ozone (O_3_) is considered to be a major threat to human health, with higher O_3_ levels brought on by climate change and human activities [2]. In the Yangtze River delta region of China, as the primary pollutant, the O_3_ levels are steadily rising and becoming a public health matter, with the proportion of O_3_ ranking first from 2017 to 2022 [3]. An epidemiological study covering 338 Chinese cities from 2015 to 2020 found that the O_3_-related health impacts increased by 94.6% and 96.5% for all-cause and respiratory disability-adjusted life years [4]. Evidence suggested that ambient O_3_ exposure was associated with an increased risk of death and various diseases including respiratory diseases [5], cardiovascular diseases [6], and neurological diseases [7]. Chen et al. showed that long-term O_3_ exposure was negatively associated with an increased risk of cardiovascular mortality, particularly ischemic heart disease [8].

Pregnant women and children are generally more vulnerable to the effects of air pollutants, and exposure to air pollutants during pregnancy has been linked to adverse birth outcomes, cognitive and motor problems [9,10]. Birth defects, including malformations, deformations, and chromosomal abnormalities, are not only the major causes of neonatal mortality but also of disability. The global prevalence of congenital birth defects is 2–3% [11]. In China, birth defects are one of the main causes of death among children under 5 years of age, which may put a huge economic burden on society. A growing body of studies has suggested that exposure to ambient O_3_ during pregnancy was associated with increased risks of birth defects [12,13]. A birth cohort study performed in Foshan, China reported that O_3_ exposure during the 1st month of pregnancy increased the risk of congenital heart defect (CHD) (odds ratio [OR], 1.03, 95% confidence interval [CI]: 0.94–1.13) [14]. Similarly, a study aimed at evaluating the health effects of O_3_ exposure on pregnant women also suggested that exposure to increased levels of O_3_ during the first trimester of pregnancy may contribute to the risk of CHDs [15]. In addition to birth defects, high levels of O_3_ exposure have also been linked to adverse birth outcomes. A study in California found that O_3_ exposure was associated with low birth weight [16]. However, the evidence regarding the association of O_3_ exposure with the risk of birth defects in pregnancy is still limited and inconsistent. A study exploring ambient air pollutants and the risk of congenital anomalies in California’s San Joaquin Valley found that O_3_ was associated with reduced odds of neural tube defects [17]. Lisa et al. suggested that ambient O_3_ exposure in early pregnancy in Texas was inversely associated with septal heart defects [18]. A comprehensive and systematic assessment of the effects of O_3_ exposure at different stages on the development of various fetal systems is still lacking.

Chongqing, located in the western region of China, faces a more severe challenge in terms of birth defects due to its unique meteorological conditions and the adjustment of the birth policy. The mountainous terrain and hot summer weather of Chongqing create favorable conditions for O_3_ formation. With the adjustment of China’s fertility policy, the demand for births has been released, and the number of newborns in Chongqing will be increased. However, studies on the relationships between O_3_ exposure and birth defects are still scarce, especially those related to preconception and first-trimester exposures. Given that maternal exposure to air pollutants does not always coincide with embryonic exposure, we aimed to evaluate for the first time the associations between ambient O_3_ exposure and the risk of birth defects, using more precise estimates of individual exposure in Chongqing, and to identify the windows of susceptibility before and throughout pregnancy.

## 2. Materials and Methods

### 2.1. Birth Defect Data

Birth defect data were collected from the Chongqing Birth Defects Monitoring Center, which included 79 hospitals, covering approximately 70% of newborns in Chongqing every year. Databases included maternal age, ethnicity, reproductive history, residence, annual income, education level, sex of the defective fetus, birth defect diagnosis, date of birth, gestational age, etc. Birth defects were diagnosed and registered according to the International Classification of Disease 10th edition (ICD-10), with the codes Q00–Q07, Q10–Q18, Q20–Q28, Q30–Q34, Q35–Q37, Q38–Q45, Q50–Q56, Q60–Q64, Q65–Q79, Q80–Q89, and Q90–Q99. The inclusion criteria for pregnant women included the following conditions: 1. Pregnancy from 1 March 2017 to 30 September 2020. 2. Gestational age ≥ 3 months. 3. The permanent residence was Chongqing City. Pregnant women without a specific address and those married to close relatives were excluded. Pregnant women with syphilis, hepatitis B infection, and HIV were also excluded (see details in Figure 1). 

### 2.2. Exposure Assessment

We sourced the O_3_ data from the China High-Resolution High-Quality Near-Surface Ozone Dataset to assess the ambient air pollutant exposure. This comprehensive dataset offers detailed daily measurements of near-surface O_3_, specifically the maximum 8 h sliding average concentrations, with a spatial resolution of 10 km. The data are accessible through the National Earth System Science Data Center [National Earth System Science Data Center, National Science and Technology Infrastructure of China (http://www.geodata.cn, accessed on 22 March 2023)]. The dataset is an amalgamation of various data sources, including O_3_ observation data, satellite remote sensing of O_3_ vertical profiles, Community Multiscale Air Quality modeling, Weather Research and Forecasting modeling, vegetation indices, nighttime light data, and demographic information [19,20].

To ascertain individual exposure levels, we employed a method that involves cross-referencing the latitude and longitude of maternal residences with the corresponding grid cells from the ChinaHighO_3_ dataset using Baidu Maps(Baidu Corporation, Beijing, China). The process involves the geospatial matching of individual case latitude and longitude coordinates to the center points of 10 km × 10 km grid cells within the O_3_ exposure dataset. The shortest distance between the grid cell center and the geographic location of the case was calculated to ensure accurate spatial attribution. Each pregnant woman’s exposure was estimated based on the daily O_3_ concentration data from the specific grid cell to which her residence was assigned. Subsequently, a monthly average exposure level was derived from a 30-day average of the assigned daily O_3_ concentrations.

### 2.3. Statistical Analysis

A time-stratified case-crossover analysis was used to assess the associations between O_3_ exposure and birth defects. Maternal exposure to environmental pollutants before pregnancy may affect the embryo or fetus, and the first trimester is a critical period for fetal organ development. The first trimester before pregnancy (the 3rd month, 2nd month, and 1st month pre-pregnant) and after pregnancy (the 1st month, 2nd month, and 3rd month post-pregnant) were used as the six phases of our study, respectively. Each of the above phases was used as an exposure period, a retrospective 1:1 control period was chosen with a 6-month interval between them, and the six control periods corresponded to the 9th, 8th, 7th, 6th, 5th, and 4th months pre-pregnant. Using a case-crossover design, in which each case serves as its control, confounders that do not change over time can be naturally controlled.

Ambient O_3_ exposure concentrations were analyzed descriptively using mean, standard deviation, and percentile. The basic profile of birth defect cases was described by counts and composition ratios. Maternal O_3_ exposure during the exposure period and the control period were compared using the conditional logistic regression model. A subgroup analysis of different classifications of birth defects was conducted according to the ICD-10. The R library survival kit and forest plot kit were used for analysis and drawing, and *p* < 0.05 was considered statistically significant.

## 3. Results

Table 1 shows the maternal and fetal characteristics of the cases. A total of 6601 cases of birth defects were diagnosed in this study; a maternal age of <35 years accounted for 89.20%. The majority of the mothers (49.86%) had a junior college/bachelor’s degree or above. Among all newborns, 56.16% were male and most (87.90%) were born during 2018–2020. The highest proportion of birth defects were congenital malformations/deformations of the musculoskeletal system, congenital malformations of the circulatory system, and congenital malformations of the eye, ear, face, and neck, accounting for 30.48%, 24.27%, and 20.01%, respectively. During the study period, the median monthly average exposure to O_3_ was 72 ug/m^3^ (changes in O_3_ concentrations are shown in Appendix A). The monthly median temperature and relative humidity were 18.8 °C and 76.5%. Table 2 lists the distribution of O_3_ exposure during different periods. The mean concentration of O_3_ was higher during the exposure periods compared to the control periods.

Table 3 shows the associations between the birth defect risk and O_3_ exposure during different periods. In general, O_3_ exposure around pregnancy was associated with increased risks of birth defects, and the most significant effect estimate was observed in the first month before pregnancy: a 10 ug/m^3^ increase of O_3_ was related to an elevation of 4.2% (95%CI: 3.4–5.1%). 

Figure 2 presents the effect modification of individual characteristics. In age-specific analyses, the significant association between birth defects and O_3_ exposure appeared in the 3 months before pregnancy for mothers ≥35 years, and the strongest positive association was shown in the second month before pregnancy (OR = 1.045, 95%CI: 1.019–1.037). Compared with others, more obvious associations between birth defects and O_3_ exposure were observed in mothers with low education during all periods. For households with different incomes, the most significant association was observed in low-income families in the second month after pregnancy (OR = 1.086, 95%CI: 1.034–1.140). Similarly, we also found that mothers from rural areas were more susceptible to O_3_ exposure, and the strongest association was in the first month after pregnancy (OR = 1.053, 95%CI: 1.034–1.072). Appendix A shows the results of the Z-tested after subgroup analyses.

The risk of developing different types of birth defects in association with O_3_ exposure varied greatly (Table 4). The associations between O_3_ exposure and congenital malformations and deformations of the musculoskeletal system (Q65–Q79) were statistically significant during almost all exposure periods. In all models, the most positive and statistically significant association between O_3_ exposure and birth defects was other congenital malformations (Q80–Q89), which was observed in the second month of pregnancy (OR = 1.11, 95%CI: 1.05–1.17). 

## 4. Discussion

In this time-stratified case-crossover study, we identified a significantly increased risk of some types of birth defects in association with maternal exposure to O_3_ in the periconceptional period. The two months before and after pregnancy are susceptible windows for exposure to O_3_, and the associations may be modified by education, household income, and residence area, which may have public health significance.

To our knowledge, this is the first study focusing on the associations between maternal exposure to O_3_ during different periods and the risk of various types of birth defects in China. 

Although many epidemiologic studies have reported associations between maternal exposure to O_3_ and birth defects, the results and the affected subtypes of birth defects have been inconsistent [21]. A comprehensive review including 27 papers explored air pollution and adverse birth outcomes in China and found that exposure to O_3_ during pregnancy may increase the risk of congenital heart diseases and other birth defects [22]. In a Texas study with more than 1.4 million births, exposure to O_3_ in early pregnancy was found to be correlated with septal heart defects [18]. However, in a similar study, estimating the odds of 26 congenital birth defect phenotypes with respect to air pollutant exposure in San Joaquin Valley, California, there were no significant associations observed [23]. Tan et al. conducted a review to investigate the epidemiological evidence of air pollutants on pregnancy outcomes during the pregnancy process and reported that the effects of air pollutants on adverse pregnancy outcomes were small or with no effect [24]. Besides O_3_ exposure, the negative effects of other air pollutants on pregnancy outcomes were also observed. A national birth defect prevention study in the United States observed modest associations between congenital limb deficiencies and air pollutants including nitrogen dioxide (NO_2_), carbon monoxide (CO), and sulfur dioxide (SO_2_) during pregnancy [25]. Zhou et al. evaluated the association between air pollutants and orofacial clefts in four U.S. states and found that PM_2_._5_ significantly increased the risk of cleft palate alone [26]. A systematic review and meta-analysis of the epidemiological literature suggested that there were statistically significant associations between increased risk-specific CHD subtypes and PM_10_, PM_2_._5_, CO, NO_2_, and O_3_ exposure [27]. The heterogeneities of these results may be because of the variation in research design, participant characteristics, exposure patterns, outcome assessment, air pollutant concentrations, and methodology. 

Although evidence on the associations between maternal exposure to air pollutants and birth defects in offspring has been growing, most studies focused on exposure during pregnancy rather than exposure during preconception [25,28,29,30,31]. Apart from trimester-specific and entire pregnancy, the preconception period may also be a susceptible exposure window. Our study appears to be the first to report that higher O_3_ levels in the two months before and after conception may increase the risk of some types of birth defects in China. In our study, not only the circulatory system, but also the congenital malformations of the musculoskeletal system, urinary system, and the five sense organs were all found to be associated with exposure to O_3_ during a critical period of embryogenesis. The development of vital organs, including cardiac, occurs at three to eight weeks of gestation, which is therefore the critical exposure window for most teratogenic agents [32]. Conventional wisdom holds that acute high-level exposure is sufficient to harm the embryo, and the associations between air pollutants and birth defects may depend not only on acute exposure but also on the buildup or accumulation of high concentrations of air pollutants or their metabolites over a longer period [33]. In this case, the risk of birth defects may increase strongly with exposure to high levels of air pollutants during the preconception period. Three months before conception is the critical time for the maturation of periovarian follicles and ovulation. Our findings suggested that high concentrations of O_3_ exposure before pregnancy should be avoided, which is consistent with the conclusion of the Chinese recommendation for pregnancy care, starting from three months before conception [34]. It is now well established that oxidative stress responses and systemic inflammation induced by O_3_ exposure are the main potential mechanisms leading to adverse birth outcomes, but the evidence for the relevant mechanism hypothesis is insufficient [35,36].

Some population characteristics may modify the health effects of air pollutants by differentiating the exposure to other risk factors and access to prevention measures [37]. In our study, we found that mothers living in areas with a lower household income were more susceptible to O_3_ exposure, which was similar to the result that socioeconomically deprived groups are more susceptible to air pollutants, reported by the World Health Organization [38]. In addition, mothers living in rural areas were more susceptible to O_3_ exposure than mothers living in urban areas, which may be related to lifestyles, and mothers in rural areas tended to do more outdoor labor. Mothers living in rural areas have lower household incomes and education, which also means they are likely to have less access to medical services, less ability to protect themselves from high levels of O_3_ exposure, and low ability to manage risks and health outcomes. There was no obvious age-modifying effect found in our study, but a nationwide surveillance-based study that examined the association between maternal exposure to air pollutants and the risk of CHDs demonstrated that mothers <35 years were more susceptible to air pollutant exposure [34]. The sensitive age range for pregnant women, and the reasons behind it, need to be further explored. While some effects were not significant in certain subgroup analyses, they may have been diluted by the association between O_3_ and other types of birth defects, which needs more studies with large samples.

Some limitations should be considered in our study. Firstly, the data were collected from a province of China, reducing the generalizability of the results to other regions. Secondly, we did not consider personal activity patterns, such as time spent indoors, which may lead to bias in the estimates of exposure levels. The misclassification of exposure may underestimate or overestimate the true effects of air pollution exposure. Thirdly, the demographic data were too sparse to control for most risk factors of birth defects, and we were unable to obtain the personal behaviors (e.g., drinking, smoking, dietary, physical activity, and the use of an air conditioner), residual confounding might exist in the models, contributing to observed differences in outcomes.

## 5. Conclusions

In summary, maternal exposure to O_3_, especially in the two months before and after pregnancy, may be positively associated with the risk of certain types of birth defects in offspring, including the musculoskeletal system, circulatory system, urinary system, as well as cervicofacial and chromosomal abnormalities. This association may be modified by education, household income, and residence area. The findings of our study add depth and clarity to the current body of evidence investigating the possible association between O_3_ and the risk of birth defects using multicenter data, which allowed for analyses of the individual types of birth defects. It may help guide efforts to prevent birth defects and highlight that protecting pregnant women from high levels of air pollutants exposure before and after pregnancy is equally important.

## Figures and Tables

**Figure 1 toxics-12-00519-f001:**
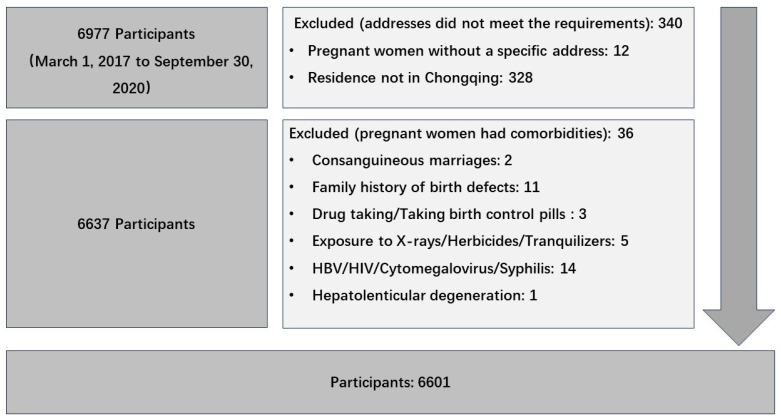
Flowchart of study participants selection process.

**Figure 2 toxics-12-00519-f002:**
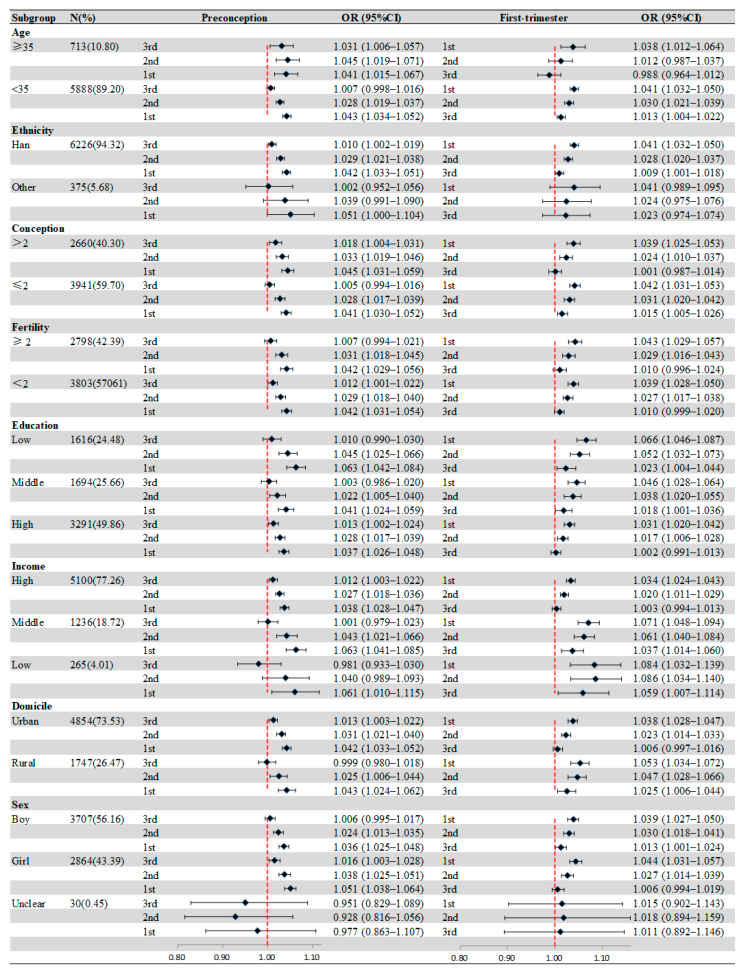
Associations of maternal O_3_ exposure with birth defects stratified by maternal or infant characteristics. Values represent ORs and the error bars are 95%CIs.

**Table 1 toxics-12-00519-t001:** Characteristics of study participants (N = 6601).

Characteristic	n	%	Characteristic	n	%
Maternal age (years)			Parity		
<35	5888	89.20	Nulliparous	3803	57.61
≥35	713	10.80	Multiparous	2798	42.39
Maternal ethnicity			Residence		
Han	6226	94.32	Urban	4854	73.53
Minority	375	5.68	Rural	1747	26.47
Birth Year			Annual household income (¥)		
2017	2	0.03	<4000	265	4.01
2018	1974	29.90	4000–7999	1236	18.72
2019	1873	28.37	≥8000	5100	77.26
2020	1955	29.62	ICD-10		
2021	797	12.07	Q65–Q79	2012	30.48
Number of pregnancy			Q20–Q28	1602	24.27
≤2	3941	59.70	Q10–Q18	1321	20.01
>2	2660	40.30	Q35–Q37	279	4.23
Maternal education			Q00–Q07	276	4.18
Less than high school	1616	24.48	Q50–Q56	269	4.08
High school	1694	25.66	Q38–Q45	222	3.36
More than high school	3291	49.86	Q90–Q99	221	3.35
Neonatal sex			Q80–Q89	190	2.88
Male	3707	56.16	Q60–Q64	172	2.61
Female	2864	43.39	Q30–Q34	37	0.56
Undivided	30	0.45	Total	6601	100.00

Note: The data within the table are categorized by ICD-10 ranges as follows: Q00–Q07, congenital malformations of the nervous system; Q10–Q18, congenital malformations of the eye, ear, face, and neck; Q20–Q28, congenital malformations of the circulatory system; Q30–Q34, congenital malformations of the respiratory system; Q35–Q37, cleft lip and palate; Q38–Q45, other congenital malformations of the digestive system; Q50–Q56, congenital malformations of reproductive organs; Q60–Q64, congenital malformations of the urinary system; Q65–Q79, congenital malformations and deformations of the musculoskeletal system; Q80–Q89, other congenital malformations; Q90–Q99, chromosomal abnormalities that cannot be classified elsewhere.

**Table 2 toxics-12-00519-t002:** O_3_ levels across the different exposure periods.

	Time Intervals (Month)	Min	Q1	Q2	Q3	Max	Mean	SD	5%	10%	90%	95%
Control periods	9th-month *	14.12	54.15	78.88	102.49	171.83	78.63	35.52	28.30	36.37	117.35	128.31
	8th-month *	13.02	51.51	75.31	101.81	163.12	76.82	35.98	28.37	35.34	117.20	128.36
	7th-month *	14.12	51.00	73.19	100.77	161.07	75.65	35.54	28.32	35.31	115.42	126.54
	6th-month *	13.65	50.66	72.71	101.32	166.46	75.91	36.00	28.41	35.58	116.37	127.17
	5th-month *	13.70	51.48	74.09	103.46	165.50	77.34	36.55	29.19	36.40	118.30	129.28
	4th-month *	13.89	53.13	77.72	104.08	168.38	78.81	36.06	30.32	37.67	118.47	129.73
Exposure periods	3rd-month *	13.89	55.48	80.28	104.50	166.18	80.28	35.44	31.29	39.68	119.27	130.14
	2nd-month *	14.42	56.08	84.46	106.00	168.05	81.85	35.60	32.04	40.90	119.88	131.27
	1st-month *	13.72	56.48	87.34	107.71	165.50	82.86	36.21	31.12	39.35	121.11	131.38
	1st-month ^#^	13.70	57.41	87.64	106.46	168.91	82.81	35.39	31.43	39.41	119.73	129.60
	2nd-month ^#^	13.49	56.94	85.53	106.21	166.15	82.20	35.50	31.72	39.75	120.02	129.35
	3rd-month ^#^	13.72	56.09	82.48	104.21	160.13	80.44	34.80	30.92	39.55	117.51	126.70

Abbreviations: SD, standard deviation. * Pre-pregnancy. ^#^ Pregnancy.

**Table 3 toxics-12-00519-t003:** The estimates of O_3_ exposure and risk of birth defects during different periods.

Exposure Periods	OR	95%CI	*p*-Value
Third month before pregnancy	1.010	1.001–1.018	0.021
Second month before pregnancy	1.030	1.021–1.038	<0.001
First month before pregnancy	1.042	1.034–1.051	<0.001
First month of pregnancy	1.041	1.032–1.049	<0.001
Second month of pregnancy	1.028	1.020–1.037	<0.001
Third month of pregnancy	1.010	1.001–1.018	0.022

**Table 4 toxics-12-00519-t004:** Risk of birth defects in offspring per increase of 10 ug/m^3^ in O_3_ level according to exposure period.

Birth Defects	Exposure Period	OR	95%CI	*p*-Value
Congenital malformations and deformations of the musculoskeletal system (Q65–Q79)	Third month before pregnancy	1.004	0.988–1.020	0.641
	Second month before pregnancy	1.035	1.018–1.051	<0.001
	First month before pregnancy	1.053	1.037–1.070	<0.001
	First month of pregnancy	1.061	1.045–1.078	<0.001
	Second month of pregnancy	1.049	1.033–1.065	<0.001
	Third month of pregnancy	1.029	1.012–1.045	<0.001
Congenital malformations of the circulatory system (Q20–Q28)	Third month before pregnancy	1.028	1.013–1.044	<0.001
	Second month before pregnancy	1.037	1.021–1.053	<0.001
	First month before pregnancy	1.034	1.018–1.050	<0.001
	First month of pregnancy	1.018	1.002–1.033	0.023
	Second month of pregnancy	1.002	0.987–1.017	0.822
	Third month of pregnancy	0.998	0.969–0.999	0.034
Congenital malformations of the eye, ear, face, and neck (Q10–Q18)	Third month before pregnancy	0.989	0.970–1.009	0.289
	Second month before pregnancy	1.016	0.996–1.035	0.116
	First month before pregnancy	1.031	1.011–1.051	0.002
	First month of pregnancy	1.035	1.015–1.055	<0.001
	Second month of pregnancy	1.039	1.019–1.060	<0.001
	Third month of pregnancy	1.025	1.005–1.046	0.016
Cleft lip and palate (Q35–Q37)	Third month before pregnancy	1.039	0.994–1.087	0.091
	Second month before pregnancy	1.042	0.999–1.086	0.053
	First month before pregnancy	1.063	1.019–1.108	0.004
	First month of pregnancy	1.063	1.018–1.110	0.006
	Second month of pregnancy	1.022	0.979–1.067	0.326
	Third month of pregnancy	0.982	0.939–1.027	0.421
Congenital malformations of the nervous system (Q00–Q07)	Third month before pregnancy	0.990	0.951–1.029	0.599
	Second month before pregnancy	0.999	0.960–1.040	0.962
	First month before pregnancy	1.029	0.987–1.072	0.175
	First month of pregnancy	1.029	0.988–1.071	0.175
	Second month of pregnancy	1.014	0.974–1.056	0.485
	Third month of pregnancy	1.014	0.974–1.057	0.493
Congenital malformations of reproductive organs (Q50–Q56)	Third month before pregnancy	1.015	0.973–1.06	0.492
	Second month before pregnancy	1.028	0.986–1.073	0.199
	First month before pregnancy	1.022	0.980–1.064	0.310
	First month of pregnancy	1.014	0.974–1.055	0.493
	Second month of pregnancy	1.003	0.963–1.044	0.899
	Third month of pregnancy	0.988	0.948–1.029	0.548
Other congenital malformations of the digestive system (Q38–Q45)	Third month before pregnancy	0.963	0.919–1.008	0.105
	Second month before pregnancy	0.988	0.944–1.034	0.607
	First month before pregnancy	1.021	0.976–1.067	0.366
	First month of pregnancy	1.046	0.999–1.096	0.056
	Second month of pregnancy	1.038	0.990–1.088	0.121
	Third month of pregnancy	1.060	1.008–1.115	0.023
Chromosomal abnormalities, cannot classified elsewhere (Q90–Q99)	Third month before pregnancy	1.060	1.016–1.105	0.007
	Second month before pregnancy	1.077	1.033–1.124	0.001
	First month before pregnancy	1.083	1.038–1.130	0.000
	First month of pregnancy	1.076	1.028–1.126	0.002
	Second month of pregnancy	1.018	0.975–1.063	0.411
	Third month of pregnancy	0.983	0.943–1.025	0.419
Other congenital malformations (Q80–Q89)	Third month before pregnancy	0.977	0.927–1.029	0.371
	Second month before pregnancy	1.027	0.976–1.080	0.308
	First month before pregnancy	1.085	1.033–1.141	0.001
	First month of pregnancy	1.134	1.073–1.199	<0.001
	Second month of pregnancy	1.110	1.053–1.169	<0.001
	Third month of pregnancy	1.074	1.022–1.130	0.005
Congenital malformations of the urinary system (Q60–Q64)	Third month before pregnancy	1.012	0.963–1.063	0.634
	Second month before pregnancy	1.036	0.986–1.089	0.163
	First month before pregnancy	1.057	1.004–1.112	0.034
	First month of pregnancy	1.026	0.978–1.078	0.297
	Second month of pregnancy	1.032	0.981–1.087	0.226
	Third month of pregnancy	1.003	0.955–1.052	0.915
Congenital malformations of the respiratory system (Q30–Q34)	Third month before pregnancy	0.983	0.879–1.101	0.771
	Second month before pregnancy	0.955	0.861–1.06	0.391
	First month before pregnancy	0.962	0.855–1.082	0.520
	First month of pregnancy	1.001	0.895–1.119	0.987
	Second month of pregnancy	0.988	0.885–1.104	0.836
	Third month of pregnancy	1.006	0.911–1.111	0.905

## Data Availability

The data of this study are available upon reasonable request to the corresponding authors.

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
