# Peer review of "Maternal Exposure to Ozone and the Risk of Birth Defects: A Time-Stratified Case-Crossover Study in Southwestern China"

_toxics, 2024, doi:10.3390/toxics12070519_

Round 1

Reviewer 1 Report

Comments and Suggestions for Authors

The authors report a novel study exploring the timing of maternal ozone exposure in relation to birth defects. Their findings indicate that exposures around the time of pregnancy (both pre- and post) are associated with increased risk of certain defects. Some clarifications and corrections are recommended prior to publication, as listed below:  

Line 27-28: What does "generalizing" mean?

Line 40: Something is missing here. "all-cause and respiratory [?]" What is the health outcome, deaths, hospitalizations?

Lines 68-69: Please explain. What meteorological conditions are challenging and what is the expected impact of the adjusted birth policy.

Lines 78-79: Please provide more explanation of Chonqing> What portion of the Chonqing population is served by the Birth Defects Monitoring Center? 

Line 92: How much spatial variability is expected with ozone? Is the 10 km grid typical?

Lines 96-97: Need references for data sources.

Line 129: Describe the birth defect type, e.g., cleft lip/palate.

Figure 1: There is a typo. Replace "Incoming" with "Income"

Line 165: "maternal or infant"

Line 170: "...during almost all exposure periods"

Line 195: Is there a word missing? "...small or without [?]"  

Line 222: with exposure (not when)

Line 252: Recommended edit: "...were too sparse to control for most risk factors"

Lines 248-254: A more complete discussion of the limitations would be useful. What are the expected influence of the limitations on the study findings? How would misclassification of exposure influence the results? 

Comments on the Quality of English Language

Minor clarifications and re-wording is recommended in some sections.

Author Response

The authors report a novel study exploring the timing of maternal ozone exposure in relation to birth defects. Their findings indicate that exposures around the time of pregnancy (both pre- and post) are associated with increased risk of certain defects. Some clarifications and corrections are recommended prior to publication, as listed below.

Response: We appreciate the evaluation of the manuscript from Reviewer 1, and we are grateful for your encouragement and suggestions, which are greatly helpful for the improvement of our manuscript.

Comments:

1) Line 27-28: What does "generalizing" mean?

Response: We thank the reviewer so much for this detail. This word is redundant and we have removed it.

2) Line 40: Something is missing here. "all-cause and respiratory [?]" What is the health outcome, deaths, hospitalizations?

Response: Thanks for your advice. We have checked the reference, the health outcome was the disability-adjusted life years. Following your suggestion, we have revised it.

3) Lines 68-69: Please explain. What meteorological conditions are challenging and what is the expected impact of the adjusted birth policy.

Response: Thanks for your helpful suggestion. The high incidence of O3 pollution generally occurs from April to September. Chongqing City has a mountainous terrain and often experiences extreme temperatures above 40°C in the summer. Strong sunshine, low clouds, weak winds, and hot weather provide more convenient conditions for O3 production. Currently, O3 has become one of the main pollutants affecting air quality in Chongqing, China. The expected impact of the adjusted birth policy refers to the fact that with the adjustment of China’s fertility policy, the demand for childbearing has been released, and the number of newborn births is likely to increase compared with the number before the policy adjustment. We have added the descriptions as “The mountainous terrain and hot summer weather of Chongqing create favorable conditions for O3 formation. With the adjustment of China’s fertility policy, the demand for births has been released, and the number of newborns in Chongqing will be increased.” in the revised paper.

4) Lines 78-79: Please provide more explanation of Chongqing> What portion of the Chongqing population is served by the Birth Defects Monitoring Center?

Response: Thanks for the suggestion. We have provided more explanation as follows: “Birth defects data were collected from the Chongqing Birth Defects Monitoring Center, which included 79 hospitals, covering approximately 70% of newborns in Chongqing every year.”

5) Line 92: How much spatial variability is expected with ozone? Is the 10 km grid typical?

Response: Thanks for your advice. According to other studies (Wei et al. Full-coverage mapping and spatiotemporal variations of ground-level ozone (O3) pollution from 2013 to 2020 across China. Remote Sens Environ. 2022; 270:112775. He et al. Marked impacts of pollution mitigation on crop yields in China. Earth’s Future. 2022, 10: e2022EF002936.), this dataset is of high quality with a cross-validation coefficient of determination (CV-R2) of 0.87, a root-mean-square error (RMSE) of ug/m3, and a mean absolute error (MAE) of 11.29 ug/m3 daily. The 10 km is grid typical, which is widely used to investigate the relationship between O3 exposure and health outcomes (Li et al. Causal Associations of Air Pollution With Cardiovascular Disease and Respiratory Diseases Among Elder Diabetic Patients. Geohealth. 2023, 7(6):e2022GH000730. Chen et al. Association of household solid fuel use and long-term exposure to ambient air pollution with estimated 10-year high cardiovascular disease risk among postmenopausal women. Environ Pollut. 2024, 342:123091.).

6) Lines 96-97: Need references for data sources.

Response: Thanks for your advice. Following your suggestion, we have added the references ([19] Wei, J.; Li, Z.; Li, K.; Dickerson, R.R.; Pinker, R.T.; Wang, J.; Liu, X.; Xue, W.; Cribb, M. Full-coverage mapping and spatiotemporal variations of ground-level ozone (O3) pollution from 2013 to 2022 across China. Remote Sens. Environ. 2022, 270, 112775. doi: 10.1016/j.rse.2021.112775. [20] He, L.; Wei, J.; Wang, Y.; Shang, Q.; Liu, J.; Yin, Y.; Frankenberg, C.; Jiang, J.H.; Li, Z.; Yung, Y.L. Marked impacts of pollution mitigation on crop yields in China. Earth’s Future. 2022, 10, e2022EF002936. doi: 10.1029/2022EF002936.).

7) Line 129: Describe the birth defect type, e.g., cleft lip/palate.

Response: Thanks for your advice. Following your suggestion, we have described the birth defect in the revised manuscript.

8) Figure 1: There is a typo. Replace "Incoming" with "Income".

Response: We thank the reviewer so much for this detail. Following your suggestion, we have replaced “incoming” with “income”.

9) Line 165: "maternal or infant"

Response: Thanks for your advice. Following your suggestion, we have revised the sentence.

10) Line 170: "...during almost all exposure periods"

Response: Thanks for your advice. Following your suggestion, we have added the word “during almost all exposure periods”.

11) Line 195: Is there a word missing? "...small or without [?]"

Response: Thanks for your advice. Following your suggestion, we have rewritten the sentence as “Tan et al. conducted a review to investigate the epidemiological evidence of air pollutants to pregnancy outcomes during the pregnancy process and reported that the effects of air pollutants on adverse pregnancy outcomes were small or with no effect.”

12) Line 222: with exposure (not when)

Response: Thanks for the suggestion. Following your advice, we have corrected the word.

13) Line 252: Recommended edit: "...were too sparse to control for most risk factors"

Response: Thanks for your advice. Following your suggestion, we have rewritten the sentence as “Thirdly, the demographic data were too sparse to control for most risk factors of birth defects...”

14) Lines 248-254: A more complete discussion of the limitations would be useful. What are the expected influence of the limitations on the study findings? How would misclassification of exposure influence the results?

Response: We fully agree with the suggestions. Following your advice, we have rewritten the limitations as follows: “Some limitations should be considered in our study. Firstly, the data were collected from a province of China, reducing the generalizability of the results to other regions. Secondly, we did not consider personal activity patterns, such as time spent indoors, which may lead to bias in estimates of exposure levels. The misclassification of exposure may underestimate or overestimate the true effects of air pollution exposure. Thirdly, the demographic data were too sparse to control for most risk factors of birth defects, and we were unable to obtain the personal behaviors (e.g., drinking, smoking, dietary, physical activity, and the use of an air conditioner), residual confounding might exist in the models, contributing to observed differences in outcomes.”

15) Comments on the Quality of English Language: Minor clarifications and re-wording is recommended in some sections.

Response: Thanks for your advice. Following your suggestion, we have further read the manuscript several times and corrected some mistakes.  

Reviewer 2 Report

Comments and Suggestions for Authors

The authors investigated the relationship between ozone exposure & the risk of birth defects. Maternal exposure to O3, especially the two months before and after pregnancy, may be positively associated with risk of certain types of birth defects in offspring, including the musculoskeletal system, circulatory system, urinary system, cervicofacial and chromosomal abnormalities. The study is very much interesting, and it focuses on a critical topic. However, there are a few areas that require clarification.

1) Which seasonal data was used for the analysis? (summer/winter) The ozone concentration may fluctuate according to the season. What is the average ozone concentration during each season?

2) How were the data normalized?

3) Do pregnant ladies have any comorbidities?

4) Authors should provide details on their ozone exposure assessment methodology. Please explain how they independently assessed the ozone exposure linked to each grid cell.

5) Please provide a graphical flow chart for the study.

Comments on the Quality of English Language

Minor language editing is needed.

Author Response

Reviewer #2:

The authors investigated the relationship between ozone exposure & the risk of birth defects. Maternal exposure to O3, especially the two months before and after pregnancy, may be positively associated with risk of certain types of birth defects in offspring, including the musculoskeletal system, circulatory system, urinary system, cervicofacial and chromosomal abnormalities. The study is very much interesting, and it focuses on a critical topic. However, there are a few areas that require clarification.

Response: We appreciate very much for your hard work to review our manuscript and your suggestions.

Comments:

1) Which seasonal data was used for the analysis? (summer/winter) The ozone concentration may fluctuate according to the season. What is the average ozone concentration during each season?

Response: We thank the reviewer so much and this is an important point. As the reviewer mentioned, the O3 concentration may fluctuate according to the season. The changes in O3 concentration have been added in Figure S1. O3 concentrations were significantly higher in summer (107.0 ug/m3) than in winter (49.2 ug/m3). The data included in our analysis were for the entire year, and we wanted to highlight that O3 exposure was strongly associated with birth defects throughout the year, not just during the summer when O3 concentrations were higher. In addition, we used a case-crossover study based on individual monthly average exposures that had been adjusted for seasonal factors.

2) How were the data normalized?

Response: Thanks for your advice. Since the O3 data was approximately normally distributed, we only considered the single exposure factor—O3, the data were not normalized.

3) Do pregnant ladies have any comorbidities?

Response: We thank the reviewer so much and this is an important point. We fully considered pregnant ladies with comorbidities in our analysis. Following your suggestions, we have added a graphical flowchart with detailed exclusion criteria. Pregnant women having other comorbidities (e.g., hypertension, diabetes, etc.) that we assessed as not affecting outcomes were not excluded.

Figure 1. Flowchart of study participants selection process.

4) Authors should provide details on their ozone exposure assessment methodology. Please explain how they independently assessed the ozone exposure linked to each grid cell.

Response: Thanks for your advice we have provided details on the O3 exposure assessment methodology “To ascertain individual exposure levels, we employed a method that involves cross-referencing the latitude and longitude of maternal residences with the corresponding grid cells from the ChinaHighO3 dataset using Baidu Maps. The process involves geospatial matching of individual case latitude and longitude coordinates to the center points of 10 km × 10 km grid cells within the O3 exposure dataset. Calculate the shortest distance between the grid cell center and the geographic location of the case to ensure accurate spatial attribution. Each pregnant woman’s exposure was estimated based on the daily O3 concentration data from the specific grid cell to which her residence was assigned. Subsequently, a monthly average exposure level was derived from a 30-day average of the assigned daily O3 concentrations.”

5) Please provide a graphical flow chart for the study.

Response: Thanks for your advice. As we mentioned in the third question,  we have added a graphical flow chart of the study (Figure 1).

6) Comments on the Quality of English Language: Minor language editing is needed.

Response: Thanks for your advice. Following your suggestion, we have further read the manuscript several times and corrected some mistakes.  

Round 2

Reviewer 2 Report

Comments and Suggestions for Authors

Authors have addressed all the comments and I have no further comments.